# Fair Deepfake Detectors Can Generalize

**Harry Cheng**
National University of Singapore
xaCheng1996@gmail.com

**Ming-Hui Liu**
Shandong University
liuminghui@mail.sdu.edu.cn

**Yangyang Guo**[*]
National University of Singapore
guoyang.eric@gmail.com

**Tianyi Wang**
National University of Singapore
terry.ai.wang@gmail.com

**Liqiang Nie**
Harbin Institute of Technology (Shenzhen)
nieliqiang@gmail.com

**Mohan Kankanhalli**
National University of Singapore
mohan@comp.nus.edu.sg

## Abstract

Deepfake detection models face two critical challenges: generalization to unseen manipulations and demographic fairness among population groups. However, existing approaches often demonstrate that these two objectives are inherently conflicting, revealing a trade-off between them. In this paper, we, for the first time, uncover and formally define a causal relationship between fairness and generalization. Building on the back-door adjustment, we show that controlling for confounders (data distribution and model capacity) enables improved generalization via fairness interventions. Motivated by this insight, we propose Demographic Attribute-insensitive Intervention Detection (DAID), a plug-and-play framework composed of: i) Demographic-aware data rebalancing, which employs inverse-propensity weighting and subgroup-wise feature normalization to neutralize distributional biases; and ii) Demographic-agnostic feature aggregation, which uses a novel alignment loss to suppress sensitive-attribute signals. Across three cross-domain benchmarks, DAID consistently achieves superior performance in both fairness and generalization compared to several state-of-the-art detectors, validating both its theoretical foundation and practical effectiveness.

## 1 Introduction

With the advancement of cutting-edge facial synthesis models, attackers can generate high-quality forged faces at minimal cost [69, 27], resulting in serious negative social implications [65]. In response to these threats, numerous deepfake detection methods have been proposed [17, 31, 80]. Employing binary real/fake classification [79, 53], these approaches have achieved promising results when trained and tested on datasets with similar distributions (*i.e.*, forged samples generated using the same manipulation techniques). However, their generalization ability remains limited when faced with previously unseen forgery methods [28, 52, 63, 39, 5, 3, 73, 19].

On the other hand, the fairness of deepfake detectors has also drawn increasing attention [13, 35]. The problem lies in that a detector should maintain consistent performance across different demographic groups, such as gender and race. However, prior studies [2, 12, 34] have predominantly shown that simply improving cross-domain generalization does not benefit all demographic subgroups equally

---

[*]Corresponding author.

39th Conference on Neural Information Processing Systems (NeurIPS 2025).

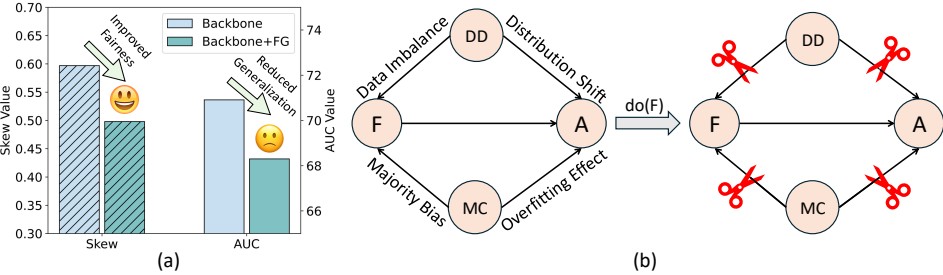

Figure 1: (a) Comparison of model performance on Celeb-DF on Skew [16] (fairness metric, the lower the better) and AUC (generalization metric, the higher the better). FG [33] is a method to improve fairness, but it may compromise the detector's generalization ability. (b) Causal graph for relationship between fairness and generalization, where data distribution ($DD$) and model capacity ($MC$) act as confounders, *i.e.*, they can affect both metrics, thereby obscuring the true causal relationship.

(*i.e.*, generalization $\not\rightarrow$ fairness). Meanwhile, as shown in Figure 1a, pushing detectors to be more fair can compromise generalizability, which arguably makes these two a trade-off [23].

Different from existing studies that treat fairness and generalization as competing objectives, our preliminary experiments show that improving detector fairness can occasionally lead to enhanced cross-domain generalization. This finding motivates our hypothesis that demographic fairness causally improves generalization performance (*i.e.*, fairness $\rightarrow$ generalization), although this effect is often obscured by confounders. To formalize this intuition, we investigate two possible confounders (data distribution ($DD$) and model capacity ($MC$)) and construct the resulting causal graph (see Figure 1b). In this graph, fairness ($F$) functions as a treatment variable exerting a causal influence on generalization ($A$). However, data distribution ($DD$) and model capacity ($MC$) act as confounders affecting both metrics and potentially obscuring the true causal relationship. Using the back-door criterion [46], which blocks spurious paths, we demonstrate a causal relationship between fairness and generalization under this causal model. Specifically, we explicitly stratify the dataset based on human demographic attributes and control for model capacity (see Section 3 for details).

To further validate our insight, we propose a novel Demographic Attribute-insensitive Intervention Detection (DAID) approach. Rather than directly optimizing for cross-domain generalization [57, 71], DAID explicitly controls for both data distribution and model capacity confounders. In doing so, DAID elucidates the causal relationship between fairness and generalization during training, and generalization can be improved by intervening on fairness. To this end, our DAID is equipped with two complementary modules. First, we apply a demographic-aware data rebalancing module, which uses adaptive sample reweighting and per-group normalization to mitigate distributional bias. Second, we propose demographic-agnostic feature aggregation, which aligns same-label samples across different demographic groups through a demographic-agnostic optimization strategy. Together, these modules serve distinct but synergistic purposes: the data rebalancing module ensures equitable representation across subgroups, while the feature aggregation module enhances the model's ability to mitigate the influence of human-related attributes. As a result, DAID effectively controls both data- and model-level confounders, while achieving substantial improvements in fairness.

We conduct extensive experiments across multiple datasets and different backbones. The results demonstrate that our approach leads to improvements in both fairness and generalization. For instance, on the DFDC [14], DFD [1], and Celeb-DF [32] datasets, our method outperforms several the state-of-the-art (SoTA) approaches. Our contributions are threefold:

- To the best of our knowledge, we are the first to establish a causal relationship where enhancing fairness leads to improved generalization in deepfake detection. This finding reveals a one-stone-hits-two-birds strategy: It enables the development of fairness-aware strategies that also enhance robustness.

- We propose a novel approach that improves generalization by promoting fairness. Our method controls the confounders, thereby isolating the causal relationship between fairness and generalization and achieving improvement in both objectives.

- We evaluate our approach on multiple datasets and backbones, showing consistent improvements in fairness and generalization. Code is provided in the supplementary materials.

## 2 Related Work

### 2.1 Deepfake Detection

**Generalization in Deepfake Detection.** Deepfake detection [21, 72, 67, 17, 37, 68, 25] is generally cast as a binary classification task. Preliminary efforts often endeavor to detect the specific manipulation traces [22, 42, 66, 77], which have shown certain improvements on intra-dataset setting. However, these methods often encounter inferior performance when applied to data with different distributions or manipulation methods. To address this generalization issue [60, 29, 51], subsequent research has increasingly devoted efforts to learning more generalized features [19, 73, 36, 57, 19]. For instance, D&L [24] introduces a novel framework that jointly leverages semantic and noise cues to achieve SOTA deepfake localization performance. RealForensics [18] exploits the visual and auditory correspondence in real videos to enhance detection performance [8]. MoE-FFD [26] represents the first innovative framework that utilizes MoE modules to achieve superior deepfake detection with significantly reduced computational cost.

**Fairness in Deepfake Detection.** Fairness in deepfake detection pertains to potential biases against certain demographic groups [62, 20, 50], particularly in terms of race and gender [44, 13]. For instance, Pu *et al.* [50] evaluate the fairness of the detector MesoInception-4 and find it to be unfair to both genders. Some recent approaches [35] have been proposed to address this problem by chasing for improved fairness metrics. For instance, Ju *et al.* [23] mitigate sharp loss landscapes during training to improve fairness within the same data domain. Lin *et al.* [33] aims to enhance cross-domain fairness by leveraging contrastive learning across different demographic subgroups. Furthermore, several approaches [56, 43] use distributionally robust optimization to improve worst-group performance, thereby addressing fairness and robustness together. Nevertheless, these methods treat fairness as the main optimization objective, without establishing a clear connection between fairness and generalization. Our DAID framework is tailored to visual deepfake data for robust cross-domain deepfake detection. Leveraging fairness as a causal intervention, DAID simultaneously boosts fairness and cross-domain robustness (generalization).

### 2.2 Causality Inference

In recent years, causal inference has emerged as a powerful tool to uncover causal relationships [4, 38, 75]. A growing body of research confirms that robust causal identification can lead to substantial improvements in model performance [40, 41, 76]. Causal inference methods can be categorized into back-door and front-door adjustment [49, 48]. The backdoor adjustment removes the confounding bias by stratifying the data according to the values of the confounders [78]. Li *et al.* [30] leverage back-door adjustment to mitigate inter- and intra-modal confounding, resulting in improved image-text matching accuracy. Chen *et al.* [7] apply back-door causal intervention to neutralize the textual bias to detect fake news. In contrast, the front door adjustment recovers the causal effect of a treatment by conditioning an observed mediator that fully carries the influence of the treatment on the outcome [6]. For instance, Zhang *et al.* [74] employ LLM-generated prompts as a mediator and calculate the causal effect between prompts and responses. In this paper, we apply back-door adjustment to block the influence of confounders, thus demonstrating the causal relationship between fairness and generalization.

## 3 Causal Analysis Between Fairness and Generalization

### 3.1 Causal Relationship Construction

**Causal Graph.** Figure 1b illustrates our assumed causal structure as a directed acyclic graph (DAG) over four variables: fairness ($F$), generalization performance ($A$), data distribution ($DD$), and model capacity ($MC$). $F$ serves as a binary treatment variable: 'low fairness' vs. 'high fairness', based on the absolute value of Skew metric (smaller Skew indicates greater fairness). $A$ is the testing-set AUC, reflecting the generalization capability. $DD$ captures the distribution of sensitive attributes (*e.g.*, race, gender), while $MC$ denotes the model's architectural capacity (*e.g.*, the number of parameters and performance on benchmarks). Since $DD$ and $MC$ influence both $F$ and $A$, we must control for them to isolate the causal effect of fairness on generalization.

This DAG contains two types of paths: i) **Causal path**: $F \to A$ represents our hypothesis that improving fairness boosts generalization; ii) **Confounding paths**: $DD \to \{F, A\}$, $MC \to \{F, A\}$, where data distribution and model capacity each affect both fairness and generalization. Confounding paths that simultaneously influence both $F$ and $A$, such as $F \leftarrow DD \to A$ and $F \leftarrow MC \to A$, can induce a *back-door effect*, introducing a spurious association between $F$ and $A$.

Therefore, it is essential to block these back-door effects for recovering the true causal effect of $F$ on $A$. To this end, we apply the **back-door adjustment** [46]. Specifically, if there exists a set of variables $\mathcal{Z}$ that satisfies the back-door criterion, we can estimate the causal relationship by conditioning on $\mathcal{Z}$.

**Definition 1 (Back-door Criterion)** *Let $\mathcal{G}$ be a causal DAG and let $X$ and $Y$ be two nodes in $\mathcal{G}$. A set of variables $\mathcal{Z}$ satisfies the* back-door criterion *relative to $X, Y$ if:*

1. *No element of $\mathcal{Z}$ is a descendant of $X \in G$.*

2. *$\mathcal{Z}$ blocks every path between $X$ and $Y$ that begins with an arrow pointing into $X$.*

In this study, $\mathcal{Z}$ is defined to include both the data and the model factors, *i.e.*, $\mathcal{Z} = \{DD, MC\}$.

**Theorem 1 (Back-door Adjustment Formula)** *If a set $\mathcal{Z}$ satisfies the back-door criterion relative to $X, Y$ in $\mathcal{G}$, then the causal effect of $X$ on $Y$ is identifiable and given by:*

$$\mathbb{P}\big(Y|do(X=x)\big) = \sum_z \mathbb{P}\big(Y|X=x, \mathcal{Z}=z\big)P(\mathcal{Z}=z). \tag{1}$$

Here, $do(X=x)$ denotes an intervention that forcibly sets $X$ to $x$, disconnecting it from its natural causes. This allows us to distinguish causal effects from spurious associations in observational data. Theorem 1 demonstrates that as long as the conditional distribution $\mathbb{P}(Y \mid X, \mathcal{Z})$ and the marginal distribution of the confounder set $\mathbb{P}(\mathcal{Z})$ can be observed, the causal effect can be identified without experimental randomization. In our context, if the influence of varying fairness levels $F$ on generalization performance $A$ remains consistent when conditioned on different values of $DD$ and $MC$, then a direct causal relationship between fairness and generalization can be established.

### 3.2 Causal Effect Estimation

According to the back-door criterion, adjusting for $\mathcal{Z} = \{DD, MC\}^2$ suffices:

$$\mathbb{P}\big(A \mid do(F=f)\big) = \sum_{dd, mc} \mathbb{P}\big(A \mid F=f, DD=dd, MC=mc\big)\,\mathbb{P}(DD=dd, MC=mc), \quad (2)$$

where $f$, $dd$, and $mc$ represent the values of $F$, $DD$, and $MC$, respectively[3]. For simplicity, we discretize the two levels of fairness with a binary variable $\{0, 1\}$, where $f = 0$ denotes low fairness. To examine the causal effect of $F$ on $A$, we define the Average Causal Effect (ACE) [55] as follows:

$$\begin{aligned} \text{ACE} &= \mathbb{P}\big(A \mid do(F=1)\big) - \mathbb{P}\big(A \mid do(F=0)\big) \\ &= \sum_{dd, mc} \Big[\mathbb{P}(A \mid F=1, dd, mc) - \mathbb{P}(A \mid F=0, dd, mc)\Big]\,\mathbb{P}(dd, mc). \end{aligned} \tag{3}$$

In other words, the causal effect is defined as the weighted average of the performance differences observed between high and low fairness conditions within each subgroup. Moreover, we define $\mu_0 = \mathbb{P}\big(A \mid do(F=0)\big)$, for any fairness level $f$, we can apply a simple substitution:

$$\begin{aligned} \mathbb{P}\big(A \mid do(F=f)\big) &= \mu_0 + f \cdot \underbrace{\Big[\mathbb{P}\big(A \mid do(F=1)\big) - \mathbb{P}\big(A \mid do(F=0)\big)\Big]}_{\text{ACE}} \\ &= \mu_0 + f \cdot \text{ACE}. \end{aligned} \tag{4}$$

This leads to a straightforward linear formulation: When $f = 0$, we have $\mathbb{P}(A \mid do(F=0)) = \mu_0$. When $f = 1$, we have $\mathbb{P}(A \mid do(F=1)) = \mu_0 + \text{ACE}$. As long as $\text{ACE} \neq 0$, we can assert that

---

[2]We approximate $\mathbb{P}(DD, MC)$ by the empirical frequency in the *held-out* test set, assuming that this set is an i.i.d. sample from the deployment population.

[3]It is worth noting that the confounding factors may vary depending on the task setting, potentially expanding beyond $DD$ and $MC$. Nevertheless, this does not affect the applicability of Equation 2.

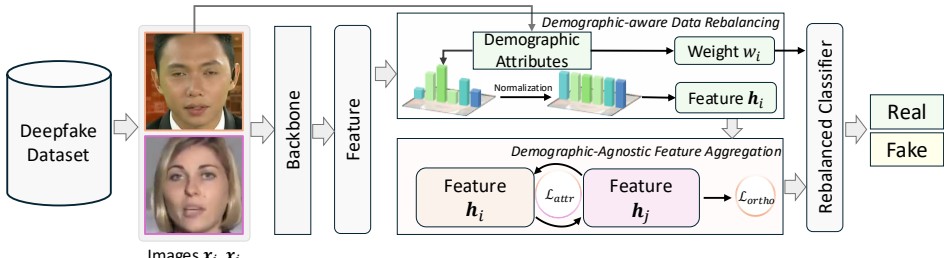

Figure 2: Overview of the proposed DAID method. **Top**: Demographic-aware Data Rebalancing. We utilize human attributes to perform demographic normalization and classifier rebalancing, which suppresses the confounding effects of $DD$. **Bottom**: Demographic-Agnostic Feature Aggregation. We introduce a demographic-agnostic loss that enhances the model's ability to filter out demographic-related information, which mitigates the confounding influence of $MC$ while improving fairness.

fairness $F$ has a causal effect on generalization performance $A$: ACE $> 0$ implies that improving fairness leads to better model performance, and ACE $< 0$ indicates the opposite.

We further design a concrete experiment to estimate the ACE to establish the causal relationship between fairness and generalization (more details are provided in the supplementary materials).

**Confounder Stratification.** For $DD$, we stratify the dataset based on the intersection of gender and race. Specifically, the dataset is first divided into two groups according to binary gender: Male and Female. Within each gender group, samples are further categorized by skin tone into three subgroups: White, Black, and Asian. Each intersection of gender and race is treated as a distinct demographic distribution. For $MC$, we employ two different architectures: Xception [54] (lower capacity) and EfficientNet [61] (higher capacity), the latter of which is known for stronger cross-domain performance [72].

**Fairness Intervention** ($do(F)$)**.** We implement two training regimes to approximate $do(F = 0)$ and $do(F = 1)$ [47]: 1) Low fairness ($F = 0$): Standard cross-entropy training. 2) High fairness ($F = 1$): Cross-entropy loss with a simple resampling strategy [9], where each sample in the cross-entropy loss is assigned a weight to suppress the over-representation of majority groups.

**ACE Estimation Results.** Based on the above procedure, we observe an average ACE gain of **2.35 percentage points** (stratified bootstrap resampling with B = 1000, $\Delta = 0.0235$, 95% CI [0.0186, 0.0280], two-sided $p < 0.001$). This result indicates that, after removing the influence of confounders, a direct relationship between fairness and generalization emerges.

### 3.3 Demographic Attribute-Insensitive Intervention Detection

Motivated by our causal findings, we conclude that, as long as confounders are properly controlled, the clear causal pathway can be leveraged to enhance generalization by intervening on more readily measurable fairness. Therefore, we introduce Demographic Attribute-Insensitive Intervention Detection (DAID), a training approach that uses fairness interventions to boost cross-domain generalization.

As illustrated in Figure 2, DAID counteracts two key confounders: data distribution ($DD$) and model capacity ($MC$) via two complementary modules: i) Demographic-aware Data Rebalancing, and ii) Demographic-Agnostic Feature Aggregation.

**Demographic-aware Data Rebalancing.** To neutralize the spurious dependency induced by the data distribution confounder $DD$, our rebalancing module includes two key components: sample-wise reweighting and representation-level normalization, that jointly calibrate both the optimization direction and the feature space geometry [45].

Firstly, we employ the inverse-probability reweighting strategy. Let $\mathbf{x}_i$ denote an input sample with sensitive demographic attributes $\mathbf{s}_i$ (*e.g.*, gender, race). To equalize the influence of majority and minority groups, we compute a sample-specific importance weight:

$$w_i = \left( \prod_{k=1}^{K} \widehat{\mathbb{P}}(\mathbf{s}_i^{(k)}) \right)^{-1},$$
(5)

where $s_i^{(k)}$ is the $k$-th sensitive attribute of $\mathbf{x}_i$, and $\widehat{\mathbb{P}}\big(s_i^{(k)}\big)$ is the empirical marginal frequency estimated from the training data. This inverse propensity weighting ensures that the expected contribution of each demographic subgroup to the loss function is approximately uniform, thus suppressing spurious correlations between $DD$ and the optimization target.

Beyond reweighting, we further mitigate $DD$-induced feature shifts by normalizing latent features within each subgroup. Denote the feature vector for $\mathbf{x}_i$ as $\mathbf{h}_i$. For each $DD$ group $dd$, we estimate the first and second moments:

$$\boldsymbol{\mu}_{dd} = \mathbb{E}_{i:dd_i=dd}[\mathbf{h}_i], \quad \boldsymbol{\sigma}^2_{dd} = \mathrm{Var}_{i:dd_i=dd}[\mathbf{h}_i], \tag{6}$$

and apply the following demographic-conditioned normalization:

$$\hat{\mathbf{h}}_i = \frac{\mathbf{h}_i - \boldsymbol{\mu}_{dd_i}}{\sqrt{\boldsymbol{\sigma}^2_{dd_i} + \varepsilon}}. \tag{7}$$

This operation aligns the group-conditioned feature distributions, removing systematic shifts induced by demographic imbalance and restoring feature comparability across subgroups.

In summary, these two strategies decouple the confounding influence of $DD$ from both model updates and representation space, yielding unbiased learning that better reflect the intrinsic relationship between fairness ($F$) and generalization ($A$).

**Demographic-Agnostic Feature Aggregation.** To eliminate the confounding influence of $MC$, we propose to encourage the model to focus on task-relevant cues while marginalizing residual demographic signals. Therefore, we perform demographic-invariant optimization in the learned representation space. The key intuition is that manipulation-consistent samples, *i.e.*, those with the same class label but differing sensitive attributes, should lead to similar internal representations.

Formally, let $\mathcal{P} = \{(\mathbf{x}_i, \mathbf{x}_j)\}$ be a set of sample pairs such that $y_i = y_j$ (same task label) and $dd_i \neq dd_j$ (different demographic attributes). We define:

$$\mathcal{L}_{\mathrm{attr}} = \frac{1}{|\mathcal{P}|} \sum_{(i,j)\in\mathcal{P}} \mathcal{L}_{\mathrm{cos}}(\hat{\mathbf{h}}_i, \hat{\mathbf{h}}_j), \tag{8}$$

where $\hat{\mathbf{h}}_i$ and $\hat{\mathbf{h}}_j$ are normalized feature vectors, and $\mathcal{L}_{\mathrm{cos}}(\cdot, \cdot)$ denotes a cosine similarity loss:

$$\mathcal{L}_{\mathrm{cos}}(\mathbf{h}_i, \mathbf{h}_j) = 1 - \cos(\mathbf{h}_i, \mathbf{h}_j) + \epsilon \tag{9}$$

where $\cos(\cdot)$ denotes the cosine similarity between feature vectors. To ensure this alignment occurs in a semantically meaningful subspace, we factorize $\hat{\mathbf{h}} \in \mathbb{R}^d$ via a low-rank projection layer:

$$\tilde{\mathbf{h}} = \mathbf{U}^\top \hat{\mathbf{h}}, \tag{10}$$

where $\mathbf{U}$ is a trainable orthonormal basis, used to filter out irrelevant directions. To avoid collapsing to trivial solutions, we regularize the projected features with:

$$\mathcal{L}_{\mathrm{ortho}} = \|\mathbf{U}\mathbf{U}^\top - \mathbf{I}\|_F^2, \tag{11}$$

where $\mathbf{I}$ is the identity matrix, and $|\cdot|_F$ denotes the Frobenius norm.

By enforcing demographic-invariant structure in a filtered representation space, this module suppresses the model's reliance on demographic features, thereby neutralizing $MC$ as a confounder and sharpening the causal interpretability of fairness-driven generalization.

**Training Objective.** We adopt a fully end-to-end optimization strategy that preserves the backbone architecture of the base detector. Specifically, we only insert our proposed modules before the classification head. It is worth noting that our approach is model-agnostic and can be seamlessly integrated into various deepfake detection backbones, which ensures inference efficiency.

Let $f_\theta : \mathbf{x} \mapsto \mathbf{h}$ denote the backbone encoder, and $g_\phi : \mathbf{h} \mapsto \hat{y}$ denote the binary classifier. Our total objective integrates the classification loss with two fairness-enhancing regularizers:

$$\mathcal{L}_{\mathrm{total}} = \mathcal{L}_{\mathrm{cls}} + \lambda_{\mathrm{attr}}\mathcal{L}_{\mathrm{attr}} + \lambda_{\mathrm{ortho}}\mathcal{L}_{\mathrm{ortho}}, \tag{12}$$

where $\mathcal{L}_{\mathrm{cls}} = \mathbb{E}(\mathbf{x}, y)\big[w_i \cdot \mathcal{B}\big(g_\phi(f_\theta(\mathbf{x})), y\big)\big]$ is the weighted binary cross-entropy loss over labels and sample-specific importance weight (see Equation (5)); $\mathcal{L}_{\mathrm{attr}}$ enforces demographic-invariant alignment between same-label samples across subgroups (see Equation (8)); $\mathcal{L}_{\mathrm{ortho}}$ ensures that the projected representation remains compact and expressive (see Equation (11)). $\lambda_{\mathrm{attr}}, \lambda_{\mathrm{ortho}}$ are hyperparameters that modulate the contribution of each loss.

| Method | DFDC | | DFD | | Celeb-DF | |
|---|---|---|---|---|---|---|
| | Skew ↓ | AUC ↑ | Skew ↓ | AUC ↑ | Skew ↓ | AUC ↑ |
| Xception [54] | 2.221 | 60.63 | 0.564 | 80.69 | 0.597 | 70.91 |
| EffcientNet [61] | 2.011 | 60.49 | 0.351 | 83.12 | 0.437 | 75.36 |
| F$^3$-Net [53] | 2.143 | 60.17 | 0.589 | 77.68 | 0.556 | 74.36 |
| Face X-ray [28] | 1.982 | 62.00 | 0.821 | 80.46 | 0.491 | 74.20 |
| SBI [57] | 2.385 | 63.39 | 0.757 | 86.43 | 0.715 | 79.76 |
| RECCE [3] | 2.622 | 61.63 | 0.738 | 80.13 | 0.644 | 70.55 |
| GRU [11] | 2.432 | 62.63 | 0.551 | 86.48 | 0.405 | 76.00 |
| CADDM [15] | 2.183 | 63.77 | 0.547 | 88.59 | 0.391 | 81.75 |
| UCF [71] | 2.272 | 60.03 | 0.510 | 81.01 | 0.619 | 71.73 |
| ProDet [10] | 2.306 | 65.89 | 0.432 | 89.18 | 0.569 | 82.71 |
| VLFFD [58] | 2.411 | 65.21 | 0.669 | 90.08 | 0.526 | 81.17 |
| ‡DAW-FDD [23] | 2.127 | 59.96 | 0.528 | 71.40 | 0.509 | 69.55 |
| ‡FG [33] | 1.932 | 60.11 | 0.447 | 80.42 | 0.498 | 68.30 |
| DAID | **1.460** | **66.85** | **0.263** | **91.15** | **0.289** | **84.39** |

Table 1: Frame-level cross-dataset performance comparison on fairness and generalization of baselines and our approach. We reproduced all baselines on three datasets and reported their Skew and AUC values. ‡: This method is proposed to enhance the fairness of the detector.

# 4 Experiments

## 4.1 Datasets and Metrics

**Datasets.** Following prior work [72, 59, 58], we employed FaceForensics++ (FF++) as the training set and evaluate the generalization performance on three other datasets: DFDC [14], DFD [1], and Celeb-DF [32]. Since none of these datasets contain native demographic annotations, we follow the data processing, annotation protocol, and sensitive attribute intersection strategy of previous fairness studies [33, 70, 23]. Specifically, we annotated each face with a combination of gender and race attributes, resulting in six demographic subgroups: Male-Asian (M-A), Male-White (M-W), Male-Black (M-B), Female-Asian (F-A), Female-White (F-W), and Female-Black (F-B).

**Metrics.** We used AUC as the primary metric to evaluate the generalizability of the model and adopted Skew as the fairness metric [16, 64, 9]. Skew is a commonly used indicator for measuring model fairness, which quantifies the performance disparity across different demographic subgroups. In our context, a lower Skew value indicates better fairness, with Skew = 0 representing perfectly fair predictions. The detailed computation of Skew is provided in the supplementary materials.

## 4.2 Implementation details

We used several deepfake detectors as backbone models, including Xception [54] ($\approx$22.9M parameters), F$^3$-Net [53] ($\approx$37.3M parameters), EfficientNet-B4 [61] ($\approx$19.3M parameters), and CADDM [15] ($\approx$21.5M parameters), to evaluate the effectiveness of DAID. Training employs AdamW (learning rate $1 \times 10^{-3}$, weight decay $4 \times 10^{-3}$) until convergence, with a batch size of 64. All input images are resized to $224 \times 224$ and normalized using ImageNet statistics. All experiments are conducted on a single NVIDIA H100 GPU.

## 4.3 Main Results

In Table 1, we reported a comparison of our method, DAID, against several SoTA baselines in terms of both fairness and generalization performance. It can be seen that DAID consistently achieves the best results in all three datasets. For instance, on Celeb-DF, our method improves fairness by 26% compared to the best-performing baseline. On the DFDC and DFD datasets, DAID achieves AUC scores of 66.85% and 91.15%, outperforming all competing methods. By controlling for confounding factors, we successfully achieve simultaneous improvements in both fairness and generalization.

It can be observed that achieving a high AUC does not necessarily imply high fairness. For example, VLFFD attains an AUC of 90.08% on the DFD dataset. However, its fairness performance lagged behind that of UCF, which exhibits significantly lower generalizability than VLFFD but demonstrates better fairness as indicated by a lower skew. Moreover, fairness-oriented methods, *i.e.*, DAW-FDD and FG, effectively enhance the fairness of the model. Nevertheless, this improvement may come

| Module | | | | Dataset | | | | | |
| --- | --- | --- | --- | --- | --- | --- | --- | --- | --- |
| Data Rebalancing | | Feature Aggregation | | DFDC | | DFD | | Celeb-DF | |
| Reweight | Normalization | $\mathcal{L}_{\mathrm{attr}}$ | $\mathcal{L}_{\mathrm{ortho}}$ | Skew ↓ | AUC ↑ | Skew ↓ | AUC ↑ | Skew ↓ | AUC ↑ |
| - | - | - | - | 2.183 | 63.77 | 0.547 | 88.59 | 0.391 | 81.75 |
| ✓ | | | | 1.719 | 64.94 | 0.295 | 89.63 | 0.340 | 83.07 |
| ✓ | ✓ | | | 1.574 | 65.96 | 0.274 | 90.67 | 0.319 | 83.98 |
| | | ✓ | | 1.750 | 65.40 | 0.273 | 89.38 | 0.327 | 83.59 |
| | | ✓ | ✓ | 1.715 | 64.96 | 0.271 | 89.55 | 0.321 | 83.88 |
| ✓ | ✓ | ✓ | | 1.495 | 66.49 | 0.266 | 91.05 | 0.292 | 84.12 |
| ✓ | ✓ | ✓ | ✓ | **1.460** | **66.85** | **0.263** | **91.15** | **0.289** | **84.39** |

Table 2: Performance of ablation studies on each module of DAID.

at the cost of reduced generalization. For instance, on the Celeb-DF dataset, FG outperforms most baselines in terms of fairness, yet its AUC score is only around 68%, significantly lower than those achieved by other methods.

## 4.4 Ablation Studies

### 4.4.1 Comparison on Modules

We reported the ablation studies on the modules of our DAID in Table 2. Specifically, we incrementally integrate each DAID module into the backbone model to assess their individual contributions. The results indicate that omitting any single module negatively impacts performance. For instance, removing the data rebalancing module, *i.e.*, no longer controlling the confounding factor $DD$, leads to a significant performance drop across all three datasets. Overall, the integration of all DAID modules yields the best performance in both generalization and fairness.

### 4.4.2 Comparison on Hyperparameters

We employ two hyperparameters, $\lambda_{\mathrm{attr}}$ and $\lambda_{\mathrm{ortho}}$, to control the relative weights of the corresponding loss functions. To investigate their impact on model generalization, we conducted a parameter sensitivity analysis, with the results shown in Figure 3. As both parameters increase, model performance initially improves and then stabilizes. Based on empirical observations, we select $\lambda_{\mathrm{attr}} = 0.7$ and $\lambda_{\mathrm{ortho}} = 0.2$ as default values. It worth noting that our method demonstrates robustness to hyperparameter selection.

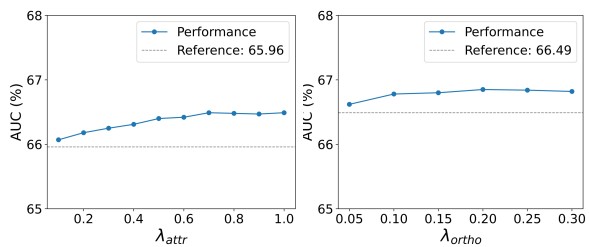

Figure 3: Hyperparameter analysis.

### 4.4.3 Comparison on Demographic Attributes

In Figure 4, we reported a radar plot that illustrates the performance of the model on the DFDC dataset at different intersections between gender and race, *e.g.*, White-Female. The left subfigure presents the AUC performance for evaluating generalization. Our DAID model outperforms the baseline across all six demographic intersections, with particularly notable improvement on the Male-Asian subgroup, where AUC increases by 30%. The right subfigure assesses fairness via the Skew metric, where our model demonstrates significantly lower skew values. This indicates that DAID achieves greater fairness in various demographic dimensions.

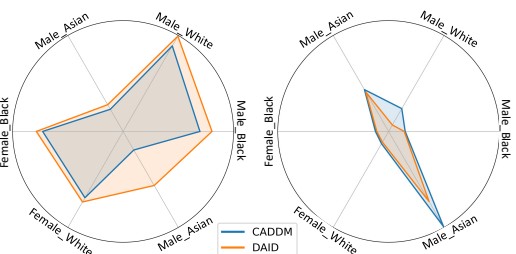

Figure 4: Radar plot for DAID. Left: AUC↑ (%) for generalization. Right: Skew↓ for fairness.

| Method | FF++ | | DFDC | | DFD | | Celeb-DF | |
|---|---|---|---|---|---|---|---|---|
| | Skew ↓ | AUC ↑ | Skew ↓ | AUC ↑ | Skew ↓ | AUC ↑ | Skew ↓ | AUC ↑ |
| Xception [54] | 0.177 | 97.85 | 2.221 | 60.63 | 0.564 | 80.69 | 0.597 | 70.91 |
| +DAID | **0.122** | **98.64** | **1.772** | **63.36** | **0.398** | **82.54** | **0.467** | **75.23** |
| EffcientNet [61] | 0.185 | 98.08 | 2.011 | 60.49 | 0.351 | 83.12 | 0.437 | 75.36 |
| +DAID | **0.136** | **98.72** | **1.697** | **63.43** | **0.264** | **84.31** | **0.352** | **78.49** |
| $F^3$-Net [53] | 0.219 | 97.32 | 2.143 | 60.17 | 0.589 | 77.68 | 0.556 | 74.36 |
| +DAID | **0.127** | **97.63** | **1.544** | **62.68** | **0.220** | **78.53** | **0.541** | **76.54** |
| CADDM [15] | 0.220 | 99.15 | 2.183 | 63.77 | 0.547 | 88.59 | 0.391 | 81.75 |
| +DAID | **0.119** | **99.26** | **1.460** | **66.85** | **0.263** | **91.15** | **0.289** | **84.39** |

Table 3: Performance comparison after applying our DAID to different backbones. All models are trained on the FF++ dataset and evaluated on four datasets. Our method consistently leads to significant improvements across all backbone architectures.

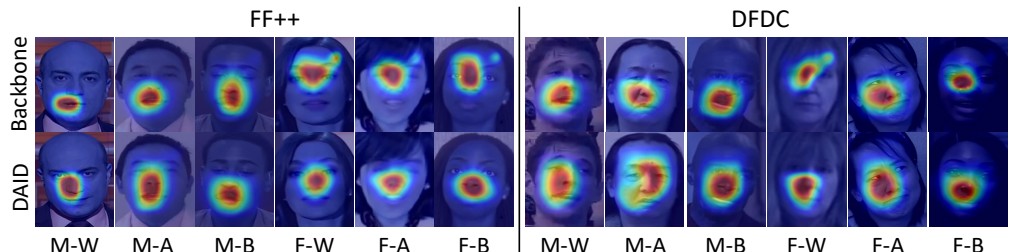

Figure 5: Non-cherry-picked Heatmaps. We included heatmaps for six demographic subgroups across two datasets: Male-Asian (M-A), Male-White (M-W), Male-Black (M-B), Female-Asian (F-A), Female-White (F-W), and Female-Black (F-B).

### 4.4.4 Comparison on Backbones

Table 3 presents the performance when applying the DAID to different backbone architectures. Specifically, we compare the performance of the four backbones, *i.e.*, Xception, EfficientNet, $F^3$-Net, and CADDM. As shown in the table, our method consistently enhances both fairness and generalization across all backbones. For instance, on Celeb-DF, applying our DAID to the Xception backbone yields a 5% increase in AUC and nearly a 20% improvement in fairness. It worth noting that this process does not require any architectural modifications to the model, leading to synergistic gains greater than the sum of individual improvements.

### 4.5 Visualization Results

In Figure 5, we present the heatmap results of the backbone model without fairness enhancement and our proposed DAID method. It can be seen that the backbone exhibits markedly different attention regions for different attributes. For instance, it focuses primarily on the lips for male subjects, while emphasizing the upper faces for female subjects. Furthermore, within the same gender, subtle differences in attention regions are also observed across different racial groups. For example, the backbone tends to focus more on the left side of the lips for the Male-White group, whereas for the Male-Black group, the nose is more frequently included in the attention region. This indicates that the backbone model conflates demographic attributes with cues for deepfake detection, potentially undermining reliable decision-making. In contrast, DAID demonstrates consistent detection patterns across both gender and race groups, effectively indicating that our method is insensitive to demographic attributes. Moreover, compared to the backbone, DAID generally focuses on broader regions of the image, reflected in its superior generalization capability.

### 4.6 Efficiency Analysis

We assess the additional computation introduced by DAID's two modules on a single NVIDIA H100 GPU (batch size 64, input resolution 224×224). For the data rebalancing module, the reweighting step adjusts only the classification loss based on subgroup frequencies, and subgroup-wise feature normalization operates directly on batch statistics. Neither requires extra gradient computations beyond standard training, resulting in negligible run-time impact. For feature aggregation module, we introduce two regularization losses and a low-rank projection layer. These involve only light

matrix multiplications and loss evaluations, resulting in minimal extra cost. On EfficientNet, standard training takes 233 min for the full session. Incorporating DAID increases this to 243 min - a relative overhead of 4.3%. Therefore, DAID's fairness-driven interventions add under 5% to total training time, making the framework practical for large-scale use.

# 5 Discussion

## 5.1 Why Fairness and Generalization May Be at Odds

The conflict between fairness and generalization may arise from both data and model characteristics. Pertaining to the data aspect, the most representative factor, *i.e.*, imbalanced distribution can lead to increased bias toward the majority group, thus improving generalization under certain limited datasets or scenarios. This however, poses the fairness problem a great challenge. Even worse, existing models tend to amplify this imbalance distribution problem, making the prediction biased. Unlike the existing methods, in this work, we propose to leverage the causal theory with the confounder controlling guidance. Informed by this, our proposed method in fact aims to rebuild balance from imbalance. Therefore, we can maintain the generalization capability of vanilla models, and can also improve the prediction fairness.

Conflicts may be difficult to resolve. For instance, in high-security systems, developers might prioritize reducing overall overall misclassification, thereby sacrificing the performance on minority groups (*e.g.*, individuals wearing unusual clothing). In contrast, for judicial models such as sentencing decisions or crime risk prediction systems, fairness across different demographic groups must be prioritized, even if it comes at the cost of reduced generalizability.

## 5.2 Connection with General Fairness Definitions

The primary fairness goal of DAID is to reduce performance disparities across demographic subgroups – in other words, to achieve a form of **group fairness**. In terms of common definitions, this aligns most with pursuing equalized performance (*e.g.*, smaller gaps in accuracy between groups), which is what the used Skew metric captures. By making feature embeddings invariant to sensitive attributes, our method mitigates the model's reliance on those attributes, thus helping to satisfy criteria related to demographic parity (outcomes independent of demographics).

## 5.3 Comprehensive General Proof for Causal Relationship

As introduced in Section 1, our causal claim regarding the relationship between fairness and generalization is defined with respect to the causal graph presented in Figure 1b. This graph is built on the assumption that data distribution ($DD$) and model capacity ($MC$) constitute an exhaustive set of confounders. If additional confounding factors are to be identified, the generality of our causal argument might be limited. Nevertheless, our experiments with DAID demonstrate that controlling for $DD$ and $MC$ is sufficient to effectively reveal a positive causal relationship between fairness and generalization. In future work, we plan to investigate whether a comprehensive and fully general proof can be established to formally substantiate this causal relationship.

# 6 Conclusion

In this paper, we demonstrate that improving fairness can causally lead to a better generalization in deepfake detection. Building on this insight, We propose the Demographic Attribute-insensitive Intervention Detection (DAID), a novel plug-and-play approach that jointly ensures demographic fairness and generalization without modifying base architectures. Extensive experiments on various benchmarks validate the theoretical foundation and practical value of DAID. Our findings reframe fairness from a mere ethical concern into a strategic lever for enhancing model robustness. By harnessing fairness as a means to improve generalization, we offer a new perspective and a practical path toward building more robust and equitable deepfake detectors. However, one limitation of our current framework is its reliance on demographic annotations. Extending DAID to operate under unlabeled or multi-dimensional fairness settings remains an important direction for future work.

## Acknowledgments and Disclosure of Funding

This document is supported by the Ministry of Education, Singapore, under its MOE AcRF TIER 3 Grant (MOE-MOET32022-0001).

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

# A    Skew Calculation

To assess fairness at the subgroup level, we introduce a **log-ratio skew metric** that quantifies the deviation between predicted and ground-truth label distributions across demographic attributes. We compute this skew separately for the `real` and `fake` classes, across both marginal (e.g., gender, race) and intersectional (e.g., female-Asian) groups.

## A.1    Definition

Given a subgroup $s \in \mathcal{S}$ and class label $c \in \{\texttt{real}, \texttt{fake}\}$, we define the skew as:

$$\text{Skew}(s, c) = \log \left( \frac{P(\hat{y} = c \mid s)}{P(y = c \mid s)} \right), \tag{13}$$

where $(P(y = c \mid s))$ denotes the empirical proportion of samples with ground-truth label $c$ in group $s$, and $(P(\hat{y} = c \mid s))$ denotes the corresponding proportion in model predictions. This skew reflects the relative distortion introduced by the model's predictions:

- $\text{Skew}(s, c) > 0$: group $s$ is overrepresented in predicted class $c$,
- $\text{Skew}(s, c) < 0$: group $s$ is underrepresented in predicted class $c$,
- $\text{Skew}(s, c) = 0$: perfect parity between prediction and ground truth.

To capture extreme disparities, we define:

$$\texttt{maxskew} = \max_{s \in \mathcal{S}, c} |\text{Skew}(s, c)|, \tag{14}$$

The `maxskew` metric corresponds to the *Skew* metric reported in our paper. It measures the degree of bias in the most skewed group, regardless of whether that group is overrepresented or underrepresented. A lower value *Skew* indicates a lower level of the most severe group bias and thus reflects a fairer model overall.

## A.2    Implementation Summary

The calculation procedure is summarized in Algorithm 1. It is applied to both marginal groups (`gender` and `race`) and intersectional groups (`gender & race`).

---
**Algorithm 1** Skew Computation for Each Demographic Group

---
**Require:** Ground-truth labels $y$, predicted probabilities $\hat{p}$, binary predictions $\hat{y}$, demographic attributes `gender`, `race`
1:  Convert predictions: $\hat{y}_i = \mathbb{I}[\hat{p}_i > 0.5]$
2:  **for** each group $s \in \mathcal{S}$ **do**
3:      **for** each class $c \in \{\texttt{real}, \texttt{fake}\}$ **do**
4:          Compute: $P(y = c \mid s)$ from ground-truth labels
5:          Compute: $P(\hat{y} = c \mid s)$ from predicted labels
6:          Compute: $\text{Skew}(s, c) = \log \left( \frac{P(\hat{y}=c|s)}{P(y=c|s)} \right)$
7:      **end for**
8:  **end for**
9:  Collect: `maxskew`, `minskew` across all $s, c$

---

We discard any subgroup $s$ for which $P(y = c \mid s) = 0$, to avoid numerical instability.

## A.3    Illustrative Example

Consider a scenario where `female-Asian` individuals constitute 10% of all `real` samples, but the model predicts 20% of `real` instances as belonging to that group. Then:

$$\text{Skew}(\texttt{female-Asian}, \texttt{real}) = \log \left( \frac{0.20}{0.10} \right) = \log(2) \approx 0.693, \tag{15}$$

indicating an overrepresentation of that subgroup in the predicted class. Conversely, a skew of $-0.693$ would indicate underrepresentation.

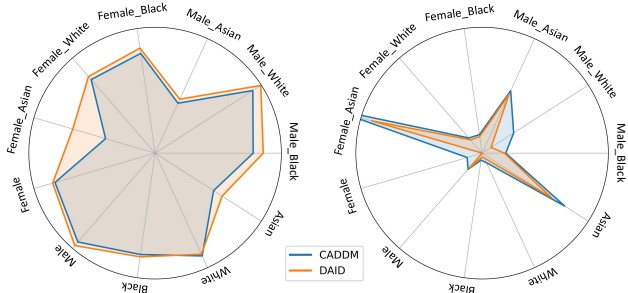

Figure 6: The performance of CADDM and DAID in terms of AUC (left) and Skew (right) across all demographic groups and their intersections.

## A.4 Application and Utility

This skew metric is used to:

- Audit group-level disparities in prediction outcomes;
- Reveal intersectional bias not captured by marginal analysis;
- Guide fairness-aware model interventions and reweighting strategies.

Figure 6 presents the fairness and generalization performance of CADDM and DAID across all groups, revealing a substantial improvement in the consistency of DAID.

## B Details for Causal Effect Estimation

### B.1 Experiment Settings

**Data Stratification (Controlling $DD$).** Following prior research [33], we apply the FF++ dataset [54] and the DFDC dataset [14] as the training set and the testing set, respectively. Both datasets are augmented with additional demographic annotations [70], *i.e.*, they contain diverse demographic attributes. In our training-testing pipeline, the entire training set is used for model training, following standard practices [63, 39]. The testing set, on the other hand, is stratified based on the intersection of gender and race. Specifically, the dataset is first partitioned based on binary gender into *Male* and *Female* groups. Within each gender group, samples are further categorized according to skin tone into three subgroups: *White*, *Black*, and *Asian*. Each intersection of gender and race is treated as a distinct distribution $dd \in DD$, with its proportion can be computed using conditional probability:

$$P(dd = (g_i, r_i)) = P(r_i|g_i) \times P(g_i), \tag{16}$$

where the $g_i$ and $r_i$ are the value of gender and race, respectively.

**Model Capacity (Controlling $MC$).** We employ two different architectures: Xception [54] and EfficientNet [61] to simulate the effect of model capacity. The latter, EfficientNet, has demonstrated superior generalization performance [72] and therefore serves as the representative of a more complex model.

**Fairness Intervention ($do(F)$).** We create two fairness levels by in-processing de-biasing [47]. Specifically, we employ standard cross-entropy training to conduct the training process with low-fairness ($f_{\text{low}}$):

$$\mathcal{L}_{\text{low}} = \frac{1}{N} \sum_{i=1}^{N} \left[ -y_i \log(\hat{y}_i) - (1 - y_i) \log(1 - \hat{y}_i) \right], \tag{17}$$

where $y_i \in \{0, 1\}$ and $\hat{y}_i \in (0, 1)$ are the ground truth and predicted labels for the $i$-th sample, respectively. Moreover, we adopt a simple resampling strategy [9], where each sample in the cross-entropy loss is assigned a weight to suppress the over-representation of majority groups, to formulate

| Method | Fairness Level | Skew | Male_Black | Male_White | Male_Asian | Female_Black | Female_White | Female_Asian |
|---|---|---|---|---|---|---|---|---|
| Xception | $f_{low}$ | 2.22 | 56.61 | 64.19 | 44.95 | 58.98 | 58.35 | 60.91 |
| | $f_{high}$ | 2.06 | 61.74 | 62.86 | 49.46 | 65.83 | 58.81 | 62.34 |
| EfficientNet | $f_{low}$ | 2.01 | 55.36 | 65.00 | 37.32 | 53.35 | 60.49 | 51.36 |
| | $f_{high}$ | 1.91 | 55.51 | 65.29 | 39.44 | 60.89 | 61.81 | 74.95 |

Table 4: Observed model AUC (%) under different data distribution ($DD$) and model capacity ($MC$). The 'Skew' column represents the fairness metric of the model, where lower values indicate better fairness.

the high-fairness ($f_{\text{high}}$) version:

$$\mathcal{L}_{\text{high}} = \sum_{i=1}^{N} (1 - \lambda_i) \left[ -y_i \log(\hat{y}_i) - (1 - y_i) \log(1 - \hat{y}_i) \right], \tag{18}$$

where $\lambda_i$ is a weighting factor defined based on the proportion of a specific group in the dataset. For example, if the subgroup with $dd = (male, white)$ constitutes 70% of the dataset, then all 'male-white' samples are assigned a weight of 0.3 during training.

**Measurement and Computation.** To estimate the causal effect of our bias intervention, we first compute the conditional AUC within each stratum defined by DD and MC factors. Specifically, for each combination $(dd, mc)$ and for each bias setting $f \in \{f_{\text{low}}, f_{\text{high}}\}$, we evaluate the empirical stratum AUC on the held-out testing dataset as

$$\hat{A}_{dd,mc}(f) = P\big(A \mid F = f, \ DD = dd, \ MC = mc\big). \tag{19}$$

This is obtained by selecting all test samples whose sensitive-attribute stratum equals $dd$ and whose model architecture equals $mc$, then measuring the fraction correctly classified under the fairness intervention $f$.

Subsequently, we compute the marginal weight of each stratum from the full test set,

$$w_{dd,mc} = P\big(DD = dd, \ MC = mc\big). \tag{20}$$

The $w_{dd,mc}$ is the proportion of test samples that fall into stratum $(dd, mc)$. We then apply the back-door adjustment formula to recover the interventional AUC under each fairness level:

$$\hat{A}(f) = \sum_{dd, \, mc} \hat{A}_{dd,mc}(f) \, w_{dd,mc}. \tag{21}$$

From Equation (21), we sum the stratum accuracies $\hat{A}_{dd,mc}(f)$ weighted by their marginal frequencies $w_{dd,mc}$ to emulate the causal effect of setting $F = f$ for the entire population.

Finally, we perform a causal comparison between the two fairness settings. Specifically, we compare $\hat{A}(f_{\text{low}})$ against $\hat{A}(f_{\text{high}})$. A statistically significant increase in $\hat{A}$ when moving from the low-fairness to the high-fairness model provides strong empirical evidence that improving fairness causally improves overall AUC.

### B.2 Numerical Illustration

Table 5 lists the observed AUCs under every combination of $DD$ and $MC$. We then aggregate each stratum using Equation (21) and obtain $A_{\text{low}} = 52.08\%$ and $A_{\text{high}} = 53.98\%$. Stratified bootstrap resampling ($B = 1000$) further shows that moving from the low-fairness to the high-fairness model yields an average gain of **2.35 percentage points** ($\Delta = 0.0235$, 95 % CI [0.0186, 0.0280], two-sided $p < 0.001$). Thus, irrespective of DD or MC, higher fairness consistently translates into better generalisation. Combining (i) the rigorously specified DAG, (ii) back-door adjustment for identification, and (iii) stratified empirical estimates under controlled bias interventions, we obtain clear, quantitative evidence that reducing model bias causally improves overall performance.

| Method | ACC ↑ | TPR ↑ | FPR ↓ |
|--------|-------|-------|-------|
| DAW-FDD | 57.89 | 60.69 | 43.33 |
| FG | 61.27 | 65.85 | 40.73 |
| DAID | 64.99 | 72.62 | 38.34 |

Table 5: Comparison of different methods on ACC, TPR, and FPR.

# C   More Experiments

## C.1   Other Metrics

We have conducted additional metrics such as Accuracy (ACC), True Positive Rate (TPR), and False Positive Rate (FPR). Our method significantly outperforms previous fairness approaches.

## C.2   Hyperparameter and Fairness

| $\lambda_{\text{attr}}$ | | $\lambda_{\text{ortho}}$ | |
|-------|-------------|-------|-------------|
| **Value** | **Performance** | **Value** | **Performance** |
| (Reference) | 1.574 | (Reference) | 1.495 |
| 0.1 | 1.559 | 0.05 | 1.483 |
| 0.2 | 1.541 | 0.1 | 1.475 |
| 0.3 | 1.529 | 0.15 | 1.464 |
| 0.4 | 1.513 | 0.2 | 1.460 |
| 0.5 | 1.509 | 0.25 | 1.462 |
| 0.6 | 1.497 | 0.3 | 1.461 |
| 0.7 | 1.495 | | |
| 0.8 | 1.496 | | |
| 0.9 | 1.498 | | |
| 1.0 | 1.497 | | |

Table 6: Ablation on $\lambda_{attr}$ and $\lambda_{ortho}$ .

We have added the results between the two hyperparameters for loss functions and skew in Table 6. Overall, improvements in fairness are generally positively correlated with improvements in generalization. The best performance is achieved when $\lambda_{attr}$ reaches 0.7 and $\lambda_{ortho}$ reaches 0.2.

