# OpenReview forum: "Fair Deepfake Detectors Can Generalize"
_NeurIPS.cc/2025/Conference — NeurIPS 2025 poster_

### Official Review · Reviewer_V3RS · 2025-06-29

**Clarity:** 4
**Significance:** 3
**Originality:** 2
**Rating:** 5
**Confidence:** 3

**Summary:**

This submission argues that when model capacity and data distribution are controlled for, improvements to the fairness of a deepfake detector lead to improvements in their generalizability. They provide theoretical arguments for this claim based on the back-door criterion, and develop a training regime which attempts to counteract these confounders. Their results show that their technique achieves SOTA fairness and generalizability.

**Questions:**

a. Am I correct in understanding that the latent normalization is just performed once, as opposed to e.g. at every hidden layer of a neural network architecture?

b. Can you explain the DD stratification going on in lines 151-155? (related to weakness 2).

c. In Fig. 3 you show the relationship between hyperparameters (meant to improve fairness) and resulting generalizability. Would you be able to provide the relationship between the hyperparameters and the resulting fairness (skew) too?

**Ethical Concerns:**

["NO or VERY MINOR ethics concerns only"]

**Final Justification:**

See author rebuttal and my response. The added table and explanation for their methodology is sufficient to demonstrate theoretical and experimental contributions. I have minor concerns about their claim of establishing a causal relationship and about confusion as to what exactly the DD confounder refers to, but I believe these can be addressed by editing of the text. As a result, I have raised my score from 4 to 5.

**Limitations:**

Yes, with the exception of potentially-unstated or understated key assumptions of their theoretical results that limit the scope of their causality claim.

**Quality:**

3

**Strengths And Weaknesses:**

Strengths:

1. The writing is very clear.

2. The proposed method achieves state of the art on both fairness and generalizability.

3. The ablation study at the end is very comprehensive, and provides an exhaustive analysis of their proposal’s contributions and impact.

Weaknesses:

I am not fully convinced of the theoretical results. In particular:

1. The backdoor criterion is defined with respect to a causal graph. Therefore, the theoretical results are built on the assumption that data distribution (DD) and model capacity (MC) are an exhaustive set of confounders. If a different researcher assumed a larger causal graph with more confounders, then the back-door adjustment formula, Eq (2), would not be correct (i.e. it would require more marginalization).

2. While the DD variable is called “data distribution”, but my understanding of lines 151-155 is that it refers to the proportion of the data set belonging to each sensitive attribute. There are many other properties of the data distribution that can impact fairness, e.g. correlation between sensitive attribute and spurious correlations such as clothing, picture setting etc. This broader definition of data distribution is implied in the latent normalization of Sec. 3.3, and if it is believed to be a confounding variable should be added to the DAG and marginalized out in the causal analysis.

3. Lines 158-161 seem to depart from the theory that precedes it. Rather than defining the do intervention as a successful flipping of F from 0 to 1, it is defined in terms of training with or without a resampling strategy. Without understanding the relationship this has to the eventual F (which is not rigorously defined in the main paper as the threshold between low and high fairness is not established) it is hard to accept the numeric conclusions of the analysis.

In conclusion, I remain unconvinced of the claimed causal relationship. I am, however, convinced of a weaker claim: when DD and MC are controlled for, improvements in fairness lead to improvements in test AUC in the setting studied. This weaker claim is enough to motivate experimental choices, but weakens some of the stated contributions.

In general, this paper is very well written and thorough in its examination of its theoretical results. I lack the confidence to assert whether the primary experiment is lacking baselines or data sets/settings, but based on the results shown the improvements to generalizability and fairness are clearly-important contributions.

Small notes:

“it worth” line 212

Lines 200-209 are a bit confusing without the context of Eq (12), which comes later. e.g. on line 200, you state “We enforce” which is a confusing construction. What is being enforced? It might be more appropriate to say “we define”, or make clear what you are enforcing wrt L_attr.



**EDIT BASED ON DISCUSSION WITH AC**

This submission has a great deal of merit but does not adequately establish the causal relationship claimed in the abstract and introduction. In particular

- The back-door criterion only establishes a causal relationship with respect to an underlying DAG, i.e. it assumes the DAG is correct. It remains unproven that there are not other confounders.
- In particular, dataset reweighting (lines 158-161) certainly has a positive effect on fairness, but it is not exactly a do intervention. It affects something upstream of fairness, and could have unexplored confounding effects.

I think the emphasis should be shifted to instead focus on the fact that, when model capacity and data distribution are controlled for, fairness-promoting interventions also improve generalization. This is a sufficient observation to motivate the experimental sections of this work.

The broad framework of causal inference and back-door criterion does not need to be scrapped to perform this change-in-emphasis; it can be done by softening some of the claims. For example, I suggest text to the effect of "we investigate two possible confounders (data distribution and model capacity) and construct the resulting causal graph. Using the back-door criterion, we demonstrate a causal relationship between fairness and generalization under this causal model". That way, the framework used in Sec 3 can be retained, but the insufficiently-substantiated claim that the authors demonstrate a (general) causal relationship between fairness and generalization is replaced by a claim that limits the scope to the DAG in question.

---

> ### Author Rebuttal · Authors · 2025-07-29
>
> We sincerely thank the reviewer for the recognition of our work. In the following, we address each of the concerns in detail.
>
> **Weakness 1 - Confounding Factors and Correction of Eq (2)**:
>
> We are very grateful for the reviewer's insightful comments. We totally agree that in a more comprehensive causal graph, one could incorporate more confounding factors (e.g., scene context, image style). In our current work, we focus on DD and MC to demonstrate that, once these two confounders are controlled, fairness positively influences generalization. As demonstrated in Section 3.2, controlling these confounders enables an estimate of the fairness→generalization effect, which is significantly positive. Specifically, we observe an improvement in AUC (≈+2.35 points **↑**) when moving from low fairness to high fairness, with statistical significance (p < 0.001). Section 4 further confirms that fairness interventions lead to consistent generalization gains under our DD and MC controls. We fully agree  that Eq. (2) can be extended to handle more confounders. We would like to introduce a unified confounder set CF, and formulate an updated equation: `P(A|do(F = f)) = ΣP(A|F=f, CF=cf)·P(CF=cf)`. We will clarify that the original Eq. (2) is simply its special case when `CF = {DD, MC}`.
>
> **We are very grateful for the reviewer’s insightful comment on the more diverse confounders, and we fully agree that this is both sensible and practically valuable.**
>
> **Weakness 2 & Question b - The Clarification of DD:**
>
> We apolpgize for this confusion, and we would like to first detail the procedure of DD stratification, and then discuss the other properties of the data distribution related to Weakness 2.
>
> **“DD stratification” means dividing the dataset into demographic subgroups** based on the sensitive attributes (e.g. gender and race) and treating each subgroup as a distinct stratum. In our implementation, we considered the intersection of *Gender* (Male, Female) and *Race* (White, Black, Asian), yielding 6 subgroups in total :
>
> - **Male-White**, **Male-Black**, **Male-Asian**
> - **Female-White**, **Female-Black**, **Female-Asian**
>
> Each of these subgroups represents a distinct portion of the overall data distribution (DD). We **condition on these subgroups when estimating the causal effect** of fairness on generalization. In other words, during our causal analysis we calculate AUC within each stratum and then aggregate . By accounting for the data distribution this way, we “neutralize” its confounding influence – any observed improvement in generalization can be attributed to improved fairness rather than differences in demographic data balance.
>
> ***(Related to Weakness 2:)***  We appreciate the reviewer’s suggestion to consider additional latent confounders beyond DD. In our formulation, DD is defined by demographic groupings. This choice follows from our motivation to estimate fairness-driven causal effects through demographic disparities, which are the most salient factors in fairness.
>
> That said, we agree that other latent properties of the data – such as image background, clothing, or picture setting – can correlate with sensitive attributes and potentially act as confounders. While these broader factors are not explicitly represented in our causal DAG, we address their influence implicitly through two mechanisms: **1) Latent feature normalization** within demographic groups, which aligns feature distributions across subgroups and thus neutralizes subgroup-specific visual patterns that co-occur with sensitive attributes (e.g., if a race group is predominantly photographed in a specific setting). **2) Feature alignment loss**, which encourages latent features of samples belonging to different subgroups but sharing the same real/fake label to be close in the feature space, thereby mitigating the model’s dependence on spurious correlations such as background-related biases. These modules functionally mitigate the effect of style- or context-related biases (even if not explicitly modeled as DAG nodes).
>
> **We appreciate the suggestion to broaden the DD definition in the DAG. Incorporating richer confounder sets would allow more fine-grained causal modeling.**
>
> **Weakness 3 - Clarification on do(F) Intervention:**
>
> We sincerely apologize for this confusion. Our do(F) intervention is intended to enhance the fairness of an initially low-fairness model, and establish a low-fairness (F=0) versus high-fairness (F=1) conditions. This is achieved by the resampling strategy, which is a widely adopted approach that has been used in prior fairness research [1] [2]. This resampling-based intervention effectively breaks the usual link between sensitive attributes and model predictions, approximating an idealized scenario of improved fairness. Specifically, in our implementation, the low-fairness setting is achieved through standard training (baseline). We then apply the resampling strategy to enhance fairness, thereby obtaining a high-fairness model. Quantitatively, this resampling reduced the Skew metric of **`Xception from 2.22 to 2.06 and that of EfficientNet from 2.01 to 1.91`**. The full set of numerical results from our causal analysis has been presented in the appendix to reinforce the robustness of our findings.
>
> [1] Social debiasing for fair multi-modal llms.
>
> [2] Enhancing Fairness through Reweighting: A Path to Attain the Sufficiency Rule
>
> **Small Notes:**
>
> We apologize for the confusion caused by these oversights. We will correct them accordingly. For instance, we will revise “We enforce” in line 200 to “We define” to better align with the subsequent introduction of L_attr.
>
> **Question a -  Latent Normalization:**
>
> We sincerely appreciate the insightful feedback, and the reviewer’s understanding is absolutely correct. The latent feature normalization** (subgroup-wise feature normalization) in our method is performed as a one-time operation on the final representation, **`not at every hidden layer`**. This design keeps the approach simple and focuses the normalization where group-wise distributional shifts manifest most – in the final embeddings – without repeatedly normalizing at each network layer (which may unnecessarily constrain the model’s internal representations).
>
> **Question c - Hyperparameter and Fairness:**
>
> We have added the results between these two hyperparameters and skew in the table below. Overall, improvements in fairness are generally positively correlated with improvements in generalization. The best performance is achieved when λ_attr reaches 0.7 and λ_ortho reaches 0.2.
>
> | λ_attr        | Performance | λ_ortho       | Performance |
> | ------------- | ----------- | ------------- | ----------- |
> | *(Reference)* | *1.574*     | *(Reference)* | *1.495*     |
> | 0.1           | 1.559       | 0.05          | 1.483       |
> | 0.2           | 1.541       | 0.1           | 1.475       |
> | 0.3           | 1.529       | 0.15          | 1.464       |
> | 0.4           | 1.513       | 0.2           | **1.460**   |
> | 0.5           | 1.509       | 0.25          | 1.462       |
> | 0.6           | 1.497       | 0.3           | 1.461       |
> | 0.7           | **1.495**   |               |             |
> | 0.8           | 1.496       |               |             |
> | 0.9           | 1.498       |               |             |
> | 1.0           | 1.497       |               |             |
>
> In conclusion, we sincerely appreciate the reviewer’s  positive perspectives of our work and will provide additional clarifications to address the concerns.

---

> > ### Comment · Reviewer_V3RS · 2025-08-02
> >
> > This response addresses my primary concerns. I have raised my score accordingly.
> >
> > I do not believe a more comprehensive set of confounders is needed, but I still believe the language employed by the authors that they "establish a causal relationship..." is misplaced. The authors establish that when DD and MC are controlled for, fairness-promoting strategies like data resampling lead to improve accuracy, which is an interesting contribution but from my understanding insufficient to fully establish a causal relationship between fairness and accuracy.
> >
> > Further, based on the authors' response, there is a slight gap in how DD is used in the theoretical results (lines 151-157) where it refers narrowly to data set sub-group balance, and how it is used in the rebalancing module (174-193) where it refers more broadly to properties of the sub-groups' data distributions. The module not only balances the data set through sample-wise reweighting; it also attempts to collapse the latent distribution for each subgroup to a single, shared distribution. The motivation for this second operation, the latent normalization, seems to be that it controls for differences in the data distribution within each subgroup, a dataset property that can be described as "data distribution" but is distinct from DD as it was examined in the theoretical results.
> >
> > Given that these concerns are related to the text and motivation, I believe them to be minor and have raised my score to a 5.

---

### Official Review · Reviewer_sqEo · 2025-06-30

**Clarity:** 2
**Significance:** 3
**Originality:** 2
**Rating:** 4
**Confidence:** 4

**Summary:**

The paper introduces a new method, namely DAID, for distinguishing between real and deepfake-generated images, while enforcing algorithmic fairness. The main contribution lies in framing the problem through a causal lens; the authors argue that the causal effect of fairness on generalization performance can be confounded by two observable confounders: (i) the distribution of sensitive attributes (e.g., gender, race) in the dataset, and (ii) the complexity of the predictive model. To address this, the paper investigates whether conditioning on these two confounders reveals a causal effect of fairness on generalization error. To this end, the proposed approach incorporates a loss function that (i) reweights data points based on group membership and (ii) includes a regularization term that enforces demographic invariance in the latent representations. The effectiveness of DAID is validated through empirical experiments on real-world datasets.

**Questions:**

Please see the weakness section.

**Ethical Concerns:**

["NO or VERY MINOR ethics concerns only"]

**Final Justification:**

The paper has a nice idea, but it seems to be a bit incremental given the large existing literature on algorithmic fairness. Notably, the method appears to be a combination of straightforward approaches from group fairness and individual fairness. Therefore, though I like the idea, I tend to keep my score 4 due to the novelty of the new method and the absence of proper theoretical guarantees.

**Limitations:**

yes.

**Paper Formatting Concerns:**

I have not noticed any major formatting issues.

**Quality:**

2

**Strengths And Weaknesses:**

**Strengths**

1. The paper is well-written and easy to read.
2. The motivation for the problem is clear.


**Weakness**

While the paper brings together several components relevant to enforcing fairness, such as reweighting samples and encouraging latent representations of similar images (real or deepfake) to be close, I am a bit concerned about the novelty. From my perspective, the primary contribution appears to lie in combining two existing strands of fairness approaches in an attempt to improve performance, rather than introducing a fundamentally new methodological insight. Below, I outline my main concerns in more detail:

1. The idea behind the proposed $L_{\rm attr}$ loss appears closely related to the FACE method introduced in [1] for enforcing individual fairness. However, the motivation for introducing this loss is not clear; to my understanding, the goal of this loss function is to ensure all latent representation of same $Y$ should be close to each other irrespective of their sensitive attributes, but it not clear what type of fairness/model complexity normalization it would bring. I recommend that the authors include a discussion situating their approach within the broader landscape of fairness definitions, such as demographic parity, equalized odds, or individual fairness. This would help make its fairness objectives more transparent.

2. The description of observations is not clear to me. What is $X$ and $Y$? Does $Y  = 1$ mean real or fake images? What are we observing?

3. The ACE estimation result is not clearly presented. It is unclear what is meant by an "ACE gain of 2.35 percentage points", does this imply that the estimated ACE is 0.0235? Additionally, the dataset used for computing the bootstrap result is not specified. The authors should clarify both the interpretation of the reported ACE value and the experimental setup used to obtain the bootstrap estimate.

4. I am also confused between $\tilde h$ in Equation (10) and $\hat h$. Which on is being used in the loss? It would be great if the authors write the loss function explicitly in terms of all the trainable parameters.

5. In the experimental section, some key evaluation metrics appear to be missing. For instance, [2] reports standard metrics such as False Positive Rate, True Positive Rate, and Accuracy, which are generally more interpretable and informative. I suggest that the authors consider including these metrics for improved clarity and comparability. Additionally, there are some discrepancies in the reported AUC scores; for example, [2] reports an AUC of 94.88 for DAW-FDD on the DFDC dataset, whereas this paper reports a significantly lower value of 59.96. Similar inconsistencies are observed for some other values. Why is this discrepancy?

[1] Two Simple Ways to Learn Individual Fairness Metrics from Data. Debarghya Mukherjee,  Mikhail Yurochkin, Moulinath Banerjee and Yuekai Sun.

[2] Improving Fairness in Deepfake Detection. Yan Ju, Shu Hu, Shan Jia, George H. Chen, and Siwei Lyu.

---

> ### Author Rebuttal · Authors · 2025-07-29
>
> We would like to express our sincere gratitude to the reviewer for the positive assessment of our work. Below, we address each of the concerns in detail.
>
> **Weakness 1 - Contributions & Fairness Objective:**
>
> We sincerely appreciate your insightful comments. We would like to first outline our contributions and then discuss the fairness objective.
>
> **(Contributions)** Our contributions can be broadly divided into two parts.  **`First`**, we employ causal graph to elucidate the relationship between fairness and generalization in deepfake detection. Building on this, we uncover the previously obscured potential for improving both of the two objectives. **`Second`,** we present DAID, a framework that leverages causal analysis to explicitly control confounding factors, thereby concurrently enhancing both the backbone model’s generalization and fairness. Empirical results demonstrate that our approach consistently outperforms baseline methods.
>
> **(Fairness objective)** The proposed demographic-agnostic alignment loss L_attr is indeed related to the notion of individual fairness – ensuring that *similar individuals receive similar outcomes*. In our context, “similar individuals” are those sharing the same *class label* (real or fake) but differing in sensitive attributes (e.g., gender or race). By encouraging the latent representations of same-class samples to be close regardless of demographic group, the proposed alignment loss pushes the model towards treating these individuals equivalently. **`We will cite the related work, such as the FACE method [1], in Section 2 and add a discussion on clarifying how our loss extends the concept to deepfake detection.`**
>
> Moreover, we will explicitly situate our approach within general fairness definitions. Our primary fairness goal is to reduce performance disparities across demographic subgroups – in other words, to achieve a form of **`group fairness`**. In terms of common definitions, this aligns most with pursuing equalized performance (e.g., smaller gaps in accuracy between groups), which is what the used Skew metric captures. By making feature embeddings *invariant* to sensitive attributes, our method mitigates the model’s reliance on those attributes, thus helping to satisfy criteria related to demographic parity (outcomes independent of demographics). In the revised paper, we will include a dedicated discussion of these fairness notions and how DAID’s objectives relate to them.
>
> [1] Two Simple Ways to Learn Individual Fairness Metrics from Data.
>
> **Weakness 2 - Meaning of X and Y, and Clarification of “Observations”:**
>
> We are sorry for the confusion. X and Y are abstract variables (i.e., `nodes`) in the causal graph. In the context of our study, i.e., the relationship between fairness and generalization, the variables X and Y are in fact F (fairness) and A (generalization), respectively. In particular, as demonstrated in Section 3.2, we define X to quantify the level of fairness intervention (i.e., whether an intervention method is employed to enhance fairness), while Y measures the model’s generalization performance (measured by AUC, which ranges from 0 to 1).
>
> In Sections 3.1 and 3.2, the term “observe” denotes examining how the generalization performance A varies across different levels of fairness intervention F, thereby establishing the interplay between fairness and generalization. The `“observational data”` refers to the dataset and evaluation metrics used in this analysis. In our experiments, we employ the DFDC dataset and report results in terms of our fairness metric (Skew) and generalization metric (AUC).
>
> **Weakness 3 - Interpretation of ACE:**
>
> We sincerely apologize for this confusion. The ACE refers to the weighted improvement in the AUC achieved by the high-fairness model over the baseline low-fairness model, where the high-fairness model is obtained by intervention. Specifically, on the DFDC dataset, we first record each model’s prediction scores and compute the AUC within each each demographic stratum (dd, mc). Aggregating these stratum-specific AUCs using empirical weights yields $A_{\text{low}}$ = 52.08\%, and $A_{\text{high}}$ = 53.98\%, implying a raw gain of 1.90 percentage points (The detailed results for each stratified group are presented in Table 4 of the appendix). To assess statistical reliability, we perform `stratified bootstrap resampling (B=1000) on the DFDC dataset`, and recompute the weighted AUC difference. The resulting mean effect is $\Delta$ = 0.0235 (i.e., 2.35 percentage points) with a 95% confidence interval of [0.0186,0.0280] and two-sided p<0.001. We will incorporate these details into Section 3.2 to clearly present the ACE definition, its computation, and its statistical significance.
>
> **Weakness 4 - Interpretation of  Equation (10):**
>
> Our sincere apologies for the confusion. In Equation (10) of the manuscript, we define $\tilde{h} = U^{\top}\hat{h}$, where $\hat{h}$ is the normalized feature vector and $U$ is a trainable orthonormal projection matrix. This $\tilde{h}$ is indeed the feature representation used in the alignment loss $L{\text{attr}}$. In other words, our demographic-agnostic alignment loss $L{\text{attr}}$ is computed on the projected features $\tilde{h}i$ and $\tilde{h}j$ (for pairs of samples with the same class label but different demographics). In the current manuscript, we acknowledge that the use of $\tilde{h}$ and $\hat{h}$ has led to confusion. To resolve this, we will reorder Equation (10) and use $\tilde{h}$ in the $L{\text{attr}}$ to avoid any ambiguity.
>
> We will also provide a clear description of all trainable parameters and variables in the loss function. In the revised version, we will list: $\theta$ (parameters of the backbone encoder), $\phi$ (parameters of the classifier), $U$ (parameters of the projection matrix in the feature aggregation module), along with definitions of $h$, $\hat{h}$, and $\tilde{h}$. The training objective can then be written explicitly as a function $L_{\text{total}}(\theta, \phi, U) = L_{\text{cls}}(\theta,\phi) + \lambda_{\text{attr}}L_{\text{attr}}(\theta, U) + \lambda_{\text{ortho}}L_{\text{ortho}}(U)$. We hope that can clarify the use of $\tilde{h}$ and loss function.
>
> **Weakness 5 - Additional Metrics and Discrepancies in AUC:**
>
> We totally agree that including these evaluation metrics can improve the clarity and completeness of the results. In the revised manuscript, we will incorporate additional metrics such as **`Accuracy, True Positive Rate (TPR), and False Positive Rate (FPR)`**. One sample table is shown below:
>
> | Method      | ACC ↑     | TPR ↑     | FPR ↓     |
> | ----------- | --------- | --------- | --------- |
> | **DAW-FDD** | 57.89     | 60.69     | 43.33     |
> | **FG**      | 61.27     | 65.85     | 40.73     |
> | **DAID**    | **64.99** | **72.62** | **38.34** |
>
> Regarding the **discrepancies in AUC scores**, this is a result of different experimental settings. In our experiments (as stated in Section 4.1), *all models are trained on FaceForensics++ (FF++)* and then tested on unseen datasets (DFDC, DFD, Celeb-DF) to assess generalization across domains. This  cross-dataset generalization scenario is a significantly more challenging setting, which naturally yields lower absolute AUC values. In contrast, the 94.88% AUC from [2] is achieved when training and testing on DFDC itself (within-dataset evaluation). `The two numbers are not directly comparable due to this difference in protocol.` We have reproduced** baseline methods in our cross-domain setup for a fair side-by-side comparison. We will explicitly clarify this in the revised manuscript to avoid confusion. For instance, we will add a note such as: *The baseline results reported in our tables are obtained under cross-dataset evaluation protocol (train on FF++ and test on other sets).”*
>
> [2] Improving Fairness in Deepfake Detection
>
> Finally, we once again extend our gratitude to the reviewer for meticulous review and recognition of our work.

---

> > ### Comment · Reviewer_sqEo · 2025-08-04
> >
> > Thank you for the clarification. This addresses some of my concerns and I intend to keep my score.

---

### Official Review · Reviewer_VvJy · 2025-06-30

**Clarity:** 3
**Significance:** 2
**Originality:** 2
**Rating:** 3
**Confidence:** 3

**Summary:**

This paper proposes a novel Demographic Attribute-insensitive Intervention Detection (DAID) approach to address the trade-off between improving fairness and generalization in Deepfake image detection. A causal relationship between fairness and generalization is assumed and established through back-door adjustment, implemented via two modules that counteract the confounders of data distribution and model capacity. Experiments are conducted on various datasets and DAID is compared against multiple baselines. The results suggest that the proposed DAID approach can improve both fairness and generalization simultaneously.

**Questions:**

(Q1) I agree that it is somewhat intuitive to assume a causal relationship from fairness to generalization, but could the authors provide more explanation and literature to support their choice of this assumption, rather than the other way around?

(Q2) The authors are encouraged to respond to the weaknesses mentioned above, especially (W1) and (W3), which will greatly affect my future decisions.

**Ethical Concerns:**

["NO or VERY MINOR ethics concerns only"]

**Final Justification:**

W3 was resovled.

W1 remains unresolved. It still lacks direct experiments and evidence that a causal relationship is created by the proposed method.

**Limitations:**

yes

**Quality:**

2

**Strengths And Weaknesses:**

Strengths:

(S1) This paper demonstrates that a direct relationship between fairness and generalization can be established after reducing the influence of confounders, as shown by a preliminary experiment in Section 3.2. Based on this finding, the proposed method looks sound to improve generalization by promoting fairness through the two proposed confounder-reduction modules, and therefore address the trade-off between fairness and generalization, as found in previous works.

(S2) The experiments are rich, and the authors provide valuable details and discussions on many aspects in the ablation studies.

(S3) In general, the paper is easy to follow and understand.

Weaknesses:

(W1) While the authors claim that "DAID elucidates the causal relationship between fairness and generalization during training," neither this causal relationship nor the confounder-reduced features are evaluated in the experiments, making it unclear whether the proposed two modules work as intended. Additional comments after the reviewers/ACs discussion stage: In my opinion, if the authors claim a specific approach leads to improved outcomes of something else, that approach itself should be directly evaluated, and evidence should be provided to show that it works as intended. In this work, while the results demonstrate great gains in model generalizability, the underlying causal relationship that is claimed to support these gains is not evaluated. The authors are encouraged to further investigate this matter, or to de-emphasize the claim about "establishing a causal relationship", to enhance the contributions of this work and to avoid any potential confusion.

(W2) The pre-processing and annotation pipeline for the Deepfake datasets used in this paper's experiments is unclear, although the authors claim that they follow previous studies (line 225). For example, four races are considered in [24] (Asian, White, Black, and Others), but only three of them are considered in this paper (Asian, White, Black). The choice of subgroups is very important in fairness works and can have a large impact on evaluating the model performance.

(W3) Table 1 and Table 2 do not report any standard deviations or statistical tests, making it unclear whether the minor number differences represent true marginal improvements or if they actually follow the same distribution. This affects the authors' claim that the proposed method outperforms the baselines.

---

> ### Author Rebuttal · Authors · 2025-07-29
>
> We thank the reviewers’ thorough evaluation very much and apologize for any confusion resulting from our oversight. Please find below our detailed responses, which address each of the reviewers’ concerns.
>
> **W1 & Q1 & Q2 - Causal Relationship Establishment and Confounder-Reduced Feature Validation**:
>
> We sincerely appreciate these insightful comments and would like to address the concerns from three perspectives:
>
> 1. **Causal relationship establishment.** Traditionally, fairness and generalization are regarded as being in a trade-off relationship. However, our preliminary experiments reveal that, in certain cases, enhancing the fairness of deepfake detectors can improve cross-domain generalization. For instance, random partitioning of the training set into distributionally distinct subsets or enhancing the backbone with additional regularization techniques or data augmentation modules can yield superior performance compared to the baseline model. This intriguing observation lead us to hypothesize that some latent factors may obscure the causal  link between fairness and generalization. Thanks to some causal analysis cases [1] [2], we find that the variability in model performance bears a striking resemblance to the influence of confounders. Therefore, we explore the interplay between fairness and generalization through the lens of causal graphs, and obtain empirical support for the notion that improving fairness can lead to enhanced generalization—provided that confounders are properly addressed (As discussed in Section 3.2).
>
>    [1] Domain Generalization via Causal Adjustment for Cross-Domain Sentiment Analysis
>
>    [2] Graph Out-of-Distribution Generalization via Causal Intervention
>
>    **The performance of The performance of `causal relationships`.** As shown in Table 1 of Section 4, once confounding effects are mitigated (e.g., using the Data Rebalancing module to control DD and the Feature Aggregation module to control MC), our method achieves improvements in both generalization and fairness compared to all baselines, including conventional generalization detectors and fairness detectors.
>
> 2. **Confounder-reduced features.** The proposed DAID approach comprises two modules that can be incorporated into the backbone model to mitigate the influence of confounders on the extracted features.
>
>    **The effectiveness of the resulting `confounder-reduced features` can be evaluated from two perspectives:**
>
>    - **The ablation results presented in Table 2** of the original manuscript demonstrate the effectiveness of our confounder-control strategy. Specifically, applying either the Data Rebalancing module to mitigate DD or the Feature Aggregation module to address MC individually yields notable improvements in both fairness and generalization. These findings validate that that each of our two modules effectively controls its respective confounder, and that combining them yields the best overall performance.
>    - **The heatmaps in Figure 5** of the original manuscript further illustrates how our method adjusts confounders. Specifically, attention maps of baseline models are biased. For instance, the baseline model focus on the lips when processing male subjects but on the upper faces for female subjects, implying reliance on gender‐related cues. In contrast, the DAID model exhibits a consistent attention distribution across all groups. This provides a validation of our proposed approach: After controlling for confounders, our method ensures that the model does not rely on sensitive-attribute proxies when making decisions. Instead, it attends to more generalizable, forgery-related features, thereby enhancing cross-domain performance.
>
> We sincerely thank the reviewer once again for their insightful comments. We will highlight these valuable insights more in the revised manuscript.
>
> **W2 - Exclusion of the “Others” Category:**
>
> We sincerely apologize for any confusion caused by this experimental choice and would like to offer the following two-fold clarification:
>
> 1. **Data Annotation and Rationale for omitting “Others.”**
>    We directly adopt the annotations from previously studies [1]. In the annotations, only three racial annotations—White, Black, and Asian—are explicitly defined. Images that do not fit these three groups are collectively labeled as “Others.” This “Others” group is both `quantitatively minor` (e.g, <5% in the FF++ dataset) and `qualitatively heterogeneous` (potentially comprising multiple smaller ethnic subgroups). Therefore, we restrict our analysis to the three racial groups to ensure that our evaluation is statistically sound.
>
> 2. **Impact on experimental results.** To ensure a fair comparison, we have `reimplemented` all baseline models. As illustrated in Figure 4 of the original manuscript, our method consistently achieves improvements across all subgroups.
>
>    [1] Analyzing fairness in deepfake detection with massively annotated databases.
>
> **W3 & Q2 – Statistical Testing:**
>
> We sincerely appreciate the insightful comment. In the final manuscript, we will augment our results with multiple independent trials to assess stability. Specifically, we will train the primary experiments five times using different random seeds and report the mean ± standard deviation across these runs. A concrete illustration of this format is provided in the table below.
>
> | Method      |    **DFDC (Skew / AUC)**     |     **DFD (Skew / AUC)**     |  **Celeb-DF (Skew / AUC)**   |
> | ----------- | :--------------------------: | :--------------------------: | :--------------------------: |
> | **CADDM**   | 2.174 ± 0.018 / 63.76 ± 0.63 | 0.563 ± 0.024 / 88.52 ± 0.31 | 0.386 ± 0.016 / 80.90 ± 0.60 |
> | **DAW-FDD** | 2.181 ± 0.077 / 59.92 ± 0.17 | 0.576 ± 0.051 / 71.16 ± 0.81 | 0.529 ± 0.065 / 69.86 ± 0.66 |
> | **FG**      | 1.930 ± 0.023 / 60.13 ± 0.35 | 0.451 ± 0.029 / 80.06 ± 0.47 | 0.502 ± 0.038 / 68.67 ± 0.59 |
> | **DAID**    | 1.458 ± 0.024 / 66.94 ± 0.27 | 0.268 ± 0.004 / 91.33 ± 0.36 | 0.293 ± 0.004 / 84.09 ± 0.40 |
>
> In summary, we sincerely thank the reviewer for these revision suggestions and apologize for the confusion we have caused. We will correct these issues and add new clarifications.

---

> > ### Comment · Reviewer_VvJy · 2025-08-04
> >
> > I thank the author's response. Some of my concerns are addressed, and I decided to keep my score.

---

### Official Review · Reviewer_4jNC · 2025-07-02

**Clarity:** 3
**Significance:** 4
**Originality:** 3
**Rating:** 5
**Confidence:** 3

**Summary:**

The paper motivates that demographic fairness and generalization to unseen data are not always at odds with each other. Based on this motivation, the paper proposes DAID to improve fairness between demographic groups and generalization to unseen data on deepfake detection models. Experiments show the effectiveness of the method.

**Questions:**

Please refer to weaknesses

**Ethical Concerns:**

["NO or VERY MINOR ethics concerns only"]

**Final Justification:**

The paper is a valuable addition to the field, discussing generalization and fairness. The authors addressed my feedback in the rebuttal, and I think the paper should be accepted

**Limitations:**

Yes

**Quality:**

3

**Strengths And Weaknesses:**

Strengths:
- I like this work. The method proposed is well explained, and useful for the deepfake detection problem. The problem being addressed is an important one.
- Experiments are thorough: Beyond basic experiments, effects of hyperparameters is shown. The comparison on demographic attributes is also interesting.

Weaknesses:
- This paper needs a discussion on why exactly fairness and generalization may be at odds with each other. Specifically, when is fairness and generalization at odds with each other? Is this a characteristic of the data, the model, or both? When can we observe that these results will hold and when is it simply not feasible to improve in both dimensions? I would like to see a detailed discussion on this.
- There are several works in the fairness and robustness literature in general, eg., [1], targeted towards tabular dataset that shows models can be more robust and fair. While robustness in [1] is considered as adversarial robustness, more adversarially robust models could improve generalization. I think its important to discuss works in fairness and robustness and draw a (qualitative) comparison.
- Minor issues in writing. Example: Line 273-"it worth noting". I also find it odd to explicitly list the venues of papers in Table 1; since the references are already there, it is not needed.

[1] Sharma, Shubham, Alan H. Gee, David Paydarfar, and Joydeep Ghosh. "Fair-n: Fair and robust neural networks for structured data." In Proceedings of the 2021 AAAI/ACM Conference on AI, Ethics, and Society, pp. 946-955. 2021.

---

> ### Author Rebuttal · Authors · 2025-07-29
>
> We would like to express our sincere gratitude to the reviewers for the valuable feedback on our work. Below, we address each of the concerns one by one.
>
> **Weakness 1 - Discussion on Fairness and Generalization May Be at Odds:**
>
> We sincerely appreciate this comment. Clarifying the inherent conflict between fairness and generalization is imperative.
>
> **When and why they are at odds.** We totally agree that this conflict may arise from both data and model characteristics. Pertaining to the data aspect, the most representative factor, i.e., imbalanced distribution can lead to increased bias toward the majority group, thus improving generalization under certain limited datasets or scenarios. This however, poses the fairness problem a great challenge. Even worse, existing `models` tend to amplify this imbalance distribution problem, making the prediction biased. Though some efforts have tried to equalize performance across demographic groups, the improved fairness often comes with degraded generalization. Unlike the existing methods, in this work, we propose to leverage the causal theory with the confounder controlling guidance. Enlightened by this, our proposed method in fact aims to `rebuild balance` from `imbalance`. Therefore, we can maintain the generalization capability of vanilla models, and can also improve the prediction fairness.
>
> **When conflicts are difficult to resolve.** The mitigation of such conflict also has its limitations. For instance, in certain highly specialized application scenarios, trade-offs between fairness and generalization may be inevitable. For instance, in high-security systems, developers might prioritize reducing overall overall misclassification, thereby sacrificing the performance on minority groups (e.g., individuals wearing unusual clothing). In contrast, for judicial models such as sentencing decisions or crime risk prediction systems, fairness across different demographic groups must be prioritized, even if it comes at the cost of reduced generalizability.
>
> We greatly appreciate the reviewer’s insightful discussion on this issue. These discussions help us strenghthen the quality and rigor of this work. We will highlight these new points in the revised manuscript.
>
> [1] Improving model fairness in image-based computer-aided diagnosis
>
> [2] The cost of fairness in classification
>
>
>
> **Weakness 2 - Related Work:**
>
> Thank you very much for highlighting related work on joint fairness and robustness. We will **cite such work**, including the tabular-data methods [1], and **expand Section 2** of the original manuscript to discuss them. These updated discussions are following:
>
> `several of these approaches, especially [1], use distributionally robust optimization to improve worst-group performance, thereby addressing fairness and robustness together. Unlike these fairness techniques that operate on tabular data, our DAID framework is tailored to visual deepfake data for robust cross-domain deepfake detection. Leveraging fairness as a causal intervention, DAID simultaneously boosts fairness and cross‐domain robustness (generalization). `
>
> `[1] Fair-n: Fair and robust neural networks for structured data.`
>
> **Weakness 3 - Minor Issues:**
>
> We sincerely apologize for these errors and will comprehensively revise all inaccuracies in the manuscript in accordance with the reviewer's comments. For instance, we will correct the typo “it worth noting” to **“it is worth noting”** on Line 273 . And we will remove venue names from Table 1 (and elsewhere) to avoid redundancy. We thank the reviewer for pointing them out.
>
> Finally, we once again extend our sincere gratitude to the reviewer for the insightful comments.

---

> > ### Comment · Reviewer_4jNC · 2025-08-01
> > **Response to rebuttal**
> >
> > Thank you for your response. It addresses my concerns, and I would like to keep my score

---

### Author Response · Authors · 2025-08-05
**Thank all reviewers.**

We would like to express our sincere gratitude to all reviewers for their thorough reading and constructive feedback, and we are committed to revising the manuscript to address every point in our rebuttal.

---

### Decision · Program_Chairs · 2025-09-17

**Decision:**

Accept (poster)

**Comment:**

This paper simultaneously tackles domain generalization and fairness in deepfake image detection from a causal perspective. It first discusses the causal effect of fairness on generalization, suggesting that improving fairness can lead to improved generalization when a quantity called average causal effective is positive. Then, it presents a new training approach RAID, which aims to learn a fair model by jointly minimizing a classification loss rebalanced between different groups and a prediction gap between individuals. Experimental results show RAID achieves the best fairness and generalization performance compared to many state-of-the-art baseline methods.

---

Overall, the work is interesting and the proposed approach is effective. Most reviewers are on the positive side.

A notable concern shared by many reviewers, including the most positive ones, is the suggested (causal) relations between fairness and generalization is not thoroughly investigated or discussed. Authors did a diligent job in defense, but the responses remain somewhat intuitive. Authors should incorporate the feedback in revision, including

-- discuss why exactly fairness and generalization may be at odds with each other (Reviewer 4jNC)

-- de-emphasize the claim that "DAID elucidates the causal relationship between fairness and generalization during training"  and related statements. Consider making a weaker claim such as that suggested by Reviewer V3RS.

-- discuss connections to studies on fairness-robustness relation (Reviewer 4jNC), and to the broader literature of algorithmic fairness, especially with group fairness and individual fairness techniques (Reviewer sqEo).

I also notice that the concept of Average Causal Effect (AVE) is also discussed in some other contexts e.g. [1,2]. Is there any relation between the AVE defined in this paper and those?

[1] Average Causal Effects From Nonrandomized Studies: A Practical Guide and Simulated Example, 2008.

[2] Disease Modeling and Public Health, Part B, Handbook of Statistics 2017.

---

Despite the concerns, reviewers are on the positive side. I think the proposed causal perspective is interesting and the proposed RAID has impressive performance. This work may serve a good step towards a new direction.